# Multiscale Attention via Wavelet Neural Operators for Vision Transformers

## Abstract

Transformers have achieved widespread success in computer vision. At their heart, there is a self-attention mechanism, an inductive bias that associates each token in the input with every other token through a weighted basis. The standard self-attention has quadratic complexity with the sequence length, which impedes its utility to long sequences appearing in high resolution vision. Recently, inspired by operator learning for PDEs, adaptive Fourier neural operators (AFNO) were introduced for high resolution attention based on global convolution that is efficiently implemented via FFT. However, the AFNO global filtering cannot well represent small and moderate scale structures that commonly appear in natural images. To leverage the coarse-to-fine scale structures we introduce a multiscale Wavelet attention (MWA) by leveraging wavelet neural operators which incurs linear complexity in the sequence size. We replace the attention in ViT with MWA and our experiments with CIFAR and Tiny-ImageNet classification demonstrate significant improvement over alternative Fourier-based attentions such as AFNO and global filter network (GFN).

## 1 Introduction

The success of transformer networks in Natural Language Processing (NLP) tasks has motivated their application to computer vision. Among the prominent advantages of transformers there is possibility of modeling long-range dependencies among the input sequence and supporting parallel processing compared to Recurrent Neural Networks (RNN). In addition, unlike Convolutional Neural Networks (CNN), they require minimal inductive biases for their design. The simple design of transformers also enables processing of multi-modality contents (such as images, video, text, and speech) by using the same processing blocks. It exhibits excellent scalability for large-size networks trained with huge datasets. These strengths have led to many improvements in vision benchmarks using transformer networks Vaswani et al. (2017); Khan et al.; Han et al. (2020).

A key component for the effectiveness of transformers is the proper mixing of tokens. Finding a good mixer is challenging because it needs to scale with the sequence size. The Self-Attention (SA) block in the vanila transformer suffers from quadratic complexity. In order to make mixing efficient, several ideas have been introduced. One recent approach is the Adaptive Fourier Neural Operator (AFNO), which aims to enhance mixing by utilizing the geometric structure of images. AFNO replaces the self-attention mechanism with a global convolution operator in the Fourier space. However, one major drawback of AFNO is that it is a global operator, which means it may overlook or miss the fine and moderate scale structures that are commonly present in natural images. This limitation can potentially hinder the model's ability to capture and understand the intricate details and patterns in the data Guibas et al. (2021); Khan et al..

To overcome the shortcomings of AFNO, one needs to effectively mix tokens at different scales Guibas et al. (2021). To this end, we propose the use of Wavelet transform, which is known as an effective multiscale representation for natural images in image processing. In order to learn a multiscale mixer, we adapt a variation of Wavelet Neural Operator (WNO) that has been studied for solving PDEs in fluid mechanics Tripura & Chakraborty (2022). We modify the design to account for high-resolution natural images with discontinuities due to objects and edge structures. After the architectural modifications, the MWA attention layer is shown in Figure 1. The input image is first transformed into the wavelet domain using two-dimensional Discrete Wavelet Transform (2D-DWT). Then, all coefficients from the last decomposition level are convolved with the

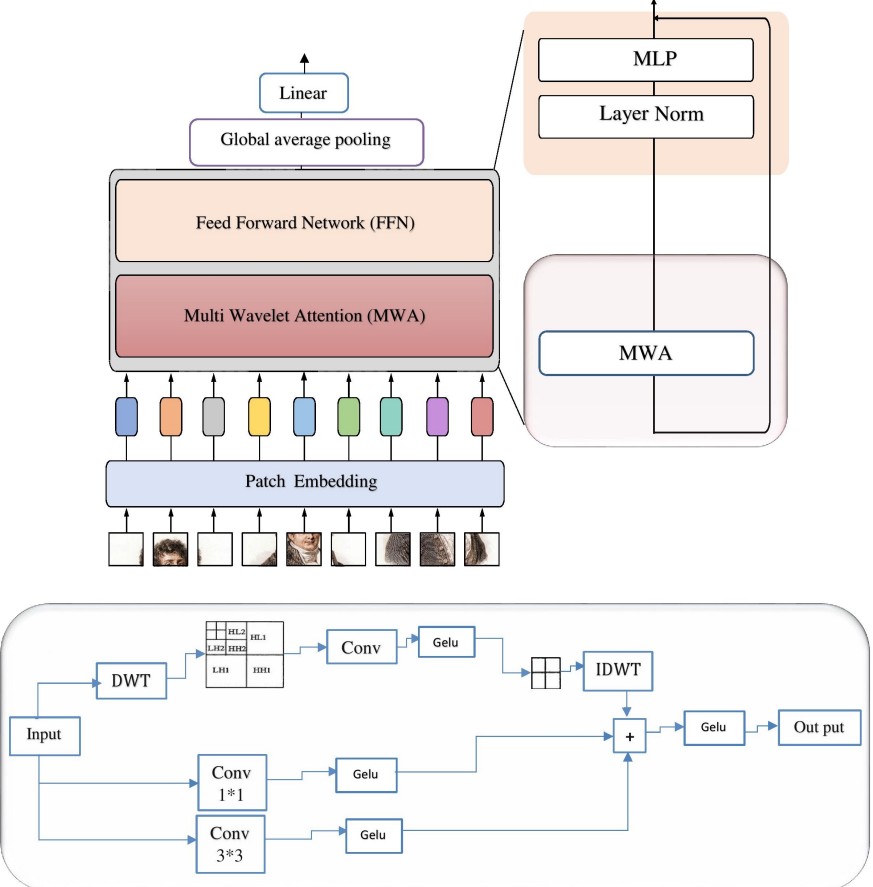

Figure 1: The general architecture of our Multiscale Wavelet Attention (MWA) for vision transformers. The bottom diagram shows the MWA architecture. Tokens are first spatially mixed using 2D-DWT. Then the tokens are filtered in the wavelet space by convolving all the coefficients of the last level of decomposition with learnable weights followed by a nonlinear GeLU activation. Then, 2D-DWT is applied to reconstruct the spatial pixel-level tokens. Weighted skip connections are also added to facilitate learning the identity map and high frequency details in the output.

learnable weights, and subsequently undergo a nonlinear GeLU activation. Then, an inverse 2D-DWT reconstructs the pixel level tokens. For 2D-DWT (and its inverse), we choose Haar Wavelet. We conducted experiments for classification, and the experiments show that our MWA has a better performance and accuracy than SA block Dosovitskiy et al. (2020) and Fourier based attentions including AFNO Guibas et al. (2021) and the Global Filter Network (GFN) Rao et al. (2021). The comparison of MWA with AFNO, GFN and SA block is mentioned in Table 1 in terms of the number of parameters and complexity.

## 2 RELATED WORKS

Several works have been introduced to improve the efficiency of the attention mechanism in transformers. We divide them into three main categories.

**Graph based attentions**. those include (1) sparse attention with fixed patterns; which reduces the attention by limiting the field-of-view to predetermined patterns such as local windows in sparse transformers Child et al. (2019); (2) sparse attention with learnable patterns; where a fixed pattern is learned from data; e.g., axial transformer Ho et al. (2019) and reformers Kitaev et al. (2020); (3) memory; another prominent method is to use a peripheral memory module that can access multiple tokens at the same time. A common form is global memory that can access the entire sequence

e.g., set transformers Lee et al. (2019); (4) low-rank methods approximate the SA via a low-rank matrix; e.g., linformers Wang et al. (2020); (5) kernel methods; use kernel trick to approximate linear transformers Katharopoulos et al. (2020); (6) recurrence is another method to improve the efficiency of the transformer e.g., compressive transformer Rae et al. (2019).

**MLP based attention**. Several works have recently been proposed that use MLP to replace self-interesting layers in feature transformation and fusion, such as ResMLP Touvron et al. (2022), which replaces layer normalization with affine transformation. A recently proposed gMLP Liu et al. (2021) also uses a spatial gating unit to reweight features in the spatial dimension. However, all models including MLP that are used to combine tokens spatially have two basic drawbacks: (1) Similar to SA, MLPs still require quadratic complexity with the sequence size; (2) MLP mixers have static weights, and they are not dynamic with respect to the input.

**Fourier based attention**. Recently, FNet, GFN, and AFNO models have been presented, which incur linear complexity. FNet Lee-Thorp et al. (2021) is an efficient transformer where each layer consists of a Fourier transform sublayer followed by a feedforward sublayer. Basically, the SA layers are replaced by Fourier transform with no learnable weights, and two-dimensional Discrete Fourier Transform (2D-DFT) is applied to embed the sequence length and hidden dimension. Another efficient transformer is the Global Filter Network (GFN), which aims to replace SA with a static global convolution filter. GFN however lacks adaptivity Rao et al. (2021). AFNO, was introduced from the operator learning perspective to solve the shortcomings of GFN, by introducing weight sharing and block diagonal structure on the learnable weights that makes it scalable Guibas et al. (2021). AFNO however, suffers from global biases and does not represent multiscale appearing commonly in natural images. The novelty of our proposed method is to account for multiscale structures using MWA attention.

**Neural operators**. Neural operators are a powerful concept in machine learning that learn the mapping between two functions in continuous space Chen & Chen (1995). They can be trained once and then used for prediction on any input function, making them highly versatile. Originally used for solving Partial Differential Equations (PDEs), neural operators have been extended to computer vision tasks by treating images as RGB-valued functions Li et al. (2020). This generalization allows us to leverage operator learning in computer vision and opens up new possibilities for solving complex vision problems. By capturing and modeling the patterns and relationships in visual data, neural operators offer a promising approach for enhancing the performance of various computer vision tasks. In this work, we adopt Wavelet neural operators that implement multiscale convolution via DWT which has been very successful for solving nonlinear and chaotic PDEs.

## 3 PRELIMINARIES AND PROBLEM STATEMENT

Consider a two-dimensional $3 \times m \times n$ RGB image $x$ that is divided into small and non-overlapping patches. After patching with patch size $p$, the image can be seen as a two-dimensional grid $3 \times h \times w$, where $h = \frac{m}{p}$ and $w = \frac{n}{p}$. Each RGB patch then undergoes a linear projection that creates tokens with a $d$-dimensional embedding, namely $d \times h \times w$. In order to preserve the position information, a $d$-dimensional position embedding is also added to each token. Since then, the transformer network processes the two-dimensional sequence of tokens by mixing them over the layers using the attention module that creates the final representation for end tasks Guibas et al. (2021); Khan et al.; Sahiner et al. (2022).

SA learns similarity among tokens Han et al. (2020). However, quadratic complexity with the sequence size hinders the training of high-resolution images Tay et al. (2022). Our goal is to replace attention with a compute and memory efficient module that is aware of the multiscale structures in the natural images for downstream tasks. Before delving into the details of multiscale attention, let us overview global-scale attention based on AFNO.

### 3.1 ADAPTIVE FOURIER NEURAL OPERATOR

In order to leverage the geometric structure of images, the AFNO, relies on convolution with a global filter that is as big as the input tokenized image. AFNO efficiently implements global convolution via FFT, which is inspired by the Fourier Neural Operator (FNO). However, FNO has a $d \times d$ weight matrix for each token ($d = \frac{n}{p}$ is the token grid dimension), so the number of parameters becomes very large for high resolution inputs. To reduce the number of parameters, AFNO imposes a block-diagonal structure on the weights. It then shares the weights among the tokens and truncates

certain frequency components using soft-thresholding and shrinkage operations Guibas et al. (2021). However, FFT can extract periodic patterns due to the use of sine and cosine functions to analyze images, but it is not suitable for studying the spatial behavior of images with non-periodic patterns. Natural images usually exhibit multiscale structures, and AFNO can miss non-periodic and small-to-medium scale structures. To model multi-scale attention and small-to-medium scale structures, our idea is to leverage wavelet transform and wavelet neural operators, which have been very successful for solving PDEs with sudden changes as discussed in the next part.

## 4 WAVELET TRANSFORM AND WAVELET NEURAL OPERATOR

### 4.1 WAVELET TRANSFORM FOR SIGNAL REPRESENTATION

Let $\psi(x) \in L^2(\mathbb{R})$ be a canonical mother wavelet that is local in both time and frequency domains. Let also $W(\Gamma)$ and $W^{-1}(\Gamma_w)$ be the forward wavelet transform and the inverse wavelet transform of an arbitrary function $\Gamma : D \to \mathbb{R}^d$. Then, the wavelet transform and the inverse are the transforms of the function $\Gamma$ with scaling and displacement parameters $\alpha \in \mathbb{R}$ and $\beta \in \mathbb{R}$. They are obtained as follows using the following integral pairs Tripura & Chakraborty (2022),

$$(W^{-1}\Gamma)(x) = \frac{1}{C_\psi} \iint_0^\infty \Gamma_w(\alpha, \beta) \frac{1}{\sqrt{|\alpha|}} \tilde{\psi}\left(\frac{x-\beta}{\alpha}\right) \frac{d\beta}{\alpha^2} d\alpha, \tag{1}$$

$$(W\Gamma)(\alpha, \beta) = \int_D \Gamma(x) \frac{1}{\sqrt{|\alpha|}} \psi\left(\frac{x-\beta}{\alpha}\right) dx, \tag{2}$$

Where $(\Gamma_w)(\alpha, \beta) = (W\Gamma)(\alpha, \beta)\psi((x-\beta)\alpha) \in L^2(R)$ is scaled and transferred to the mother wavelet. By scaling and shifting, the desired wavelets can be obtained from the mother wavelet. Each set of wavelet functions forms an orthogonal set of basis functions. Note that the term $C_\psi$ is the admissible constant which ranges in $0 \le C_\psi \le \infty$. The expression for $C_\psi$ is given as follows:

$$C\psi = 2\pi \int D \frac{|\psi(\omega)|^2}{|\omega|} d\omega \tag{3}$$

In signal representation theory, wavelet decomposition has proven successful in compressible representation with a smaller number of basis functions compared with Fourier transform. This comes from the nature of wavelet bases that can well represent trends, breakpoints, and discontinuities in higher derivatives and similarities Wirsing (2020). We aim to rely on the spatial and frequency localization power of wavelets to learn the relationship between tokens and thus learn the multiscale patterns at the internal layers of transformers. Considering these features, we adapt the WNO which has been very successful for solving nonlinear and chaotic PDEs, and we discuss in the next section Tripura & Chakraborty (2022); Graps (1995); Slimani et al. (2016).

### 4.2 WAVELET NEURAL OPERATORS

The class of shift-equivariant kernels has a notable property of being decomposable into linear combinations of eigenfunctions Soliman & Srinath (1990). Wavelet transform bases, which are a powerful class of eigenfunctions, exhibit the convolution theorem, where multiscale convolution in the spatial domain is equivalent to multiplication in the wavelet transform domain. Leveraging this property, we can now introduce the definition of Wavelet Neural Operators (WNO) Tripura & Chakraborty (2022).

**Definition (Kernel integral operator).** The kernel integral operator $K$ is defined as follows:

$$K(x)(s) = \int_D k(s;t)x(t)dt; \quad s \in D \tag{4}$$

with a continuous kernel function $k : D \times D \to \mathbb{R}^{d \times d}$. For the special case of Green's kernel, $k(s;t)$ can be expressed as $k(s;t) = k(s-t)$, and the integral of Eq.(4) leads to multiscale convolution defined below.

Table 1: Complexity, parameter count, and interpretation for MWA, AFNO, GFN and SA. $N = hw$, $d$ and $K$ refer to the sequence size, channel size, and block count in AFNO. Also, $k_1$, $k_2$ are kernel size for MWA, and $g_1$, $g_2$ are the number of groups, respectively.

| Models | Complexity (FLOPs) | Parameter Count | Interpretation |
|--------|--------------------|-----------------|----------------|
| SA | $N^2 d + 3Nd^2$ | $3d^2$ | Graph Global Conv |
| GFN | $Nd + N \log N$ | $Nd$ | Depthwise Global Conv |
| AFNO | $\frac{Nd^2}{k} + N \log N$ | $(1 + \frac{4}{k})d^2 + 4d$ | Adaptive Global Conv |
| MWA | $2mk_1 \frac{Nd^2}{g_1} + 2mk_2 \frac{Nd^2}{g_2}$ | $(\frac{k_1}{g_1} + \frac{k_2}{g_2})d^2$ | Multi Scale Conv |

**Definition (Multiscale convolution kernel operator).** Assuming that $k(s,t) = k(s-t)$, the kernel integral of Eq.(4) is rewritten as follows:

$$K(x)(s) = \int_D k(s-t)x(t)dt; \quad s \in D \tag{5}$$

The Green's kernel possesses a valuable regularization effect, enabling it to capture multiscale interactions effectively. Furthermore, it can be utilized to implement multiscale convolution efficiently through the Discrete Wavelet Transform (DWT).

**Definition (Wavelet neural operator).** For the continuous input $x \in D$, kernel $k$, and the kernel integral at token $s$, the wavelet neural operator is defined as follows:

$$K(x)(s) = W^{-1}\big(W(x) \cdot W(k)\big)(s); \quad s \in D \tag{6}$$

Here, $\cdot$ denotes matrix multiplication, and $W$ and $W^{-1}$ represent the forward DWT and the inverse DWT.

## 5 MULTISCALE WAVELET ATTENTION

Inspired by WNO, for RGB images, our idea is to combine the tokens using DWT. We make fundamental modifications to adapt the WNO operator to images to account for high-resolution natural images with object-induced discontinuities and edge structures (images with high details). In the proposed MWA, we leverage the efficiency and effectiveness of the DWT for combining tokens. The DWT offers fast implementations and takes advantage of GPU support Barina Kucis et al. (2014). In MWA, images are converted into high-frequency and low-frequency components using DWT. In essence, high-frequency components represent edges in the image, while low-frequency components represent smooth regions. According to Figure 1, the first branch in the two-dimensional array calculates four components as follows: the approximation component (LL) that represents low-frequency components; the detail components that account for high frequencies such as horizontal (HL), vertical (LH), and diagonal (HH). In this work, we use all the coefficients of the last decomposition level.

In DWT, we transform the mother wavelet to calculate the wavelet coefficients on scales with powers of two. In this case, the wavelet $\psi(x)$ is defined as follows Tripura & Chakraborty (2022):

$$\psi_{m,t}(x) = \frac{1}{\sqrt{2^m}} \psi\left(\frac{x - t2^m}{2^m}\right) \tag{7}$$

Where the parameters $m$ and $t$ are the scaling and shifting parameters and the forward DWT wavelet transform is shown below Tripura & Chakraborty (2022):

$$W\Gamma(m,t) = \frac{1}{\sqrt{2^m}} \int_D \Gamma(x)\psi\left(\frac{x - t2^m}{2^m}\right)dx \qquad (8)$$

By fixing the scale parameter $m$ to a certain integer and shifting $t$, the DWT coefficients at level $m$ can be obtained. The DWT is implemented as a filter bank, which consists of low-pass and high-pass filters. The image is decomposed into details and approximate coefficients by passing it through these filters. The low-pass and high-pass filters, represented by $r(n)$ and $s(n)$, respectively, perform convolutions of the form $z_{\text{high}}(n) = (x * s)(n)$ and $z_{\text{low}}(n) = (x * r)(n)$, where $n$ is the number of discretization points. The detail coefficients $z_{\text{high}}(n)$ are preserved, while the approximate coefficients $z_{\text{low}}(n)$ are recursively filtered by passing them through low-pass and high-pass filters until the total number of decomposition levels is exhausted Zhang et al. (1995); Meyer (1990); Tripura & Chakraborty (2022). At each level, the length of the image is halved due to conjugate symmetry.

The general architecture of the MWA model is shown in Figure 1. The model takes non-overlapping $h \times w$ grid patches as input and projects each patch into a $d$-dimensional space. The input token tensor is defined as $x \in \mathbb{R}^{h \times d \times w}$, and the weight tensor is defined as $w \in \mathbb{R}^{(\frac{h \times w}{2^m}) \times d \times d}$ for parameterization of the kernel. MWA performs a sequence of operations for each token $(m, n) \in [w] \times [h]$, which will be discussed below.

**First step**: Unlike AFNO, which combines tokens with Discrete Fourier Transform (DFT), MWA combines tokens representing different spatial locations using DWT as

$$z_{m,n} = [DWT(x)]_{m,n} \qquad (9)$$

In DWT, the wavelet coefficients in the highest scale include the important features of the input, and only from the wavelet coefficients with the highest scale, a parameterization space with limited dimensions is obtained with information preservation. In general, the length of the wavelet coefficients is also influenced by the number of vanishing moments of the orthogonal mother wavelet. Therefore, we use coefficients $z_{m,n}$ at the highest level of decomposition.

**Second step**: While AFNO uses the multiplication between the learnable weight tensor and the coefficients obtained from the DFT, we use the convolution between a learnable weight tensor and coefficients of the last level of decomposition as follows:

$$\tilde{z}_{m,n} = z_{m,n} * w_{m,n} \qquad (10)$$

**Third step**: Unlike AFNO, which uses Inverse Discrete Fourier Transform (IDFT) to recover tokens after mixing, we use Inverse Discrete Wavelet Transform (IDWT) to update and separate tokens by using:

$$y_{m,n} = [IDWT(\tilde{z})]_{m,n} \qquad (11)$$

Using DWT, we can generate fine image details as well as the rough approximation of the image. Note, DWT and IDWT are well supported by CPU and GPU, so the proposed model has good performance on hardware.

**Fourth step**: weighted skip connections are added using two convolution layers with different kernel sizes (second and third branches of Figure 1). These convolution layers facilitate learning the identity mapping and have been proven useful for learning high frequency details.

In general, the architectural highlights are as follows:

- Network parameters are learned in the wavelet space, which are localized both in frequency and spatial domains, and thus they can learn multiscale patterns in images effectively.

- WNO is adopted from continuous PDEs and modified for discrete images by adding more nonlinearity Activation and adding convolutional skip connections. Also, both the approximation and detail coefficients of the wavelet transform are used to model the attention. This comprehensive utilization of wavelet coefficients enables a more accurate representation

of the image's important details at various scales. By combining these modifications, our approach offers a powerful framework for effectively analyzing and understanding the complexities of discrete images.

- WNO is adopted from continuous PDEs and modified for discrete images by adding more nonlinearity Activation and adding convolutional skip connections. Also, both the approximation and detail coefficients of the wavelet transform are used to model the attention.This comprehensive utilization of wavelet coefficients enables a more accurate representation of the image's important details at various scales. By combining these modifications, our approach offers a powerful framework for effectively analyzing and understanding the complexities of discrete images.

- Our model is more flexible than SA because both DWT and IDWT have no learnable parameters and can process sequences with arbitrary length.

## 6 COMPLEXITY

In this section we quantify the operation count for the proposed MWA attention. For DWT, the input is simultaneously decomposed using a low-pass filter $r(n)$ and a high-pass filter $s(n)$. In case of Haar Wavelet, the high-pass and low-pass filters have a fixed length, each of which perform $z_{high}(n) = (r * x)(n)$ and $z_{low}(n) = (s * x)(n)$, which has a complexity of $O(N)$ for the sequence size $N$. DWT also uses these two filters for decomposition. Thus, the implementation of DWT filter bank has complexity of $O(N)$ Wirsing (2020). Decomposing the input using a wavelet with level $m$ results in an image of length $n/2^m$. The convolution of the analyzed coefficients of the last level and the weights has a complexity of $O(KNd^2m/g)$ (in our proposed architecture, the level of analysis is $m = 1$). The decomposition level and number of groups plays an important role in increasing the speed of our proposed architecture. The input convolution and weights with the kernel size $k$ and the number of groups $g$ also have complexity $O(kNd^2/g)$ Wei et al.. The overall complexity of the architecture is shown in Table 1.

## 7 EXPERIMENTS

We conduct experiments to confirm the effectiveness of MWA and compare the results with different Fourier based transformers. We perform our experiment on CIFAR and Tiny-ImageNet datasets as widely used small and medium-scale benchmarks for image classification. More extensive experiments and ablations can be found in Appendix A.

**Datasets.** As mentioned, we adopt CIFAR and Tiny-ImageNet datasets. CIFAR-10 contains 60,000 images from 10 class categories, while CIFAR-100 contains 60,000 from 100 class categories. Tiny-ImageNet also contains 100,000 images with 200 classes. We report the accuracy on test data.

**Comparisons.** We compare our method with the attention block in the main transformer and the AFNO and GFN Fourier transform methods, which have similar FLOPs and number of parameters, and we see that our method can clearly perform well in small and medium sized data such as CIFAR and Tiny-ImageNet (see Table 2, Table 3 and Table 4). One of the problems with transformers is that they require a lot of data for training, and they perform poorly on medium and low data, but our method can perform better than previous transformers on small datasets.

### 7.1 ARCHITECTURE AND TRAINING

The proposed MWA block consists of three major components. The first component converts the input image taken from the previous layer into a wavelet domain using 2D-DWT (horizontal, vertical and diagonal approximation coefficients and details). Then convolution is performed on all the approximate coefficients and details of the last level of decomposition and learnable weights, which then undergo GeLU nonlinear activation. Then, an inverse 2D-DWT reconstructs the pixel-level tokens. For 2D-DWT and its inverse, we choose Haar Wavelet with decomposition level $m = 1$. For skip connections we use two-dimensional convolution with a different kernel sizes $1 \times 1$ and $3 \times 3$, followed by nonlinear GeLU activation. Finally, all three branches are gathered and passed through a non-linear GeLU activation.

We use the ViT-XS/4, ViT-S/4 and ViT-B/4 configuration for experimenting on CIFAR10-100 and Tiny-ImageNet datasets. The ViT-XS/4 configuration has 5 layers and a hidden size of 384. The ViT-S/4 configuration has 12 layers and a hidden size of 384. The ViT-B/4 configuration has 12

Table 2: Comparisons of different transformer-style architectures for image classification on CIFAR-100. All our models are trained on 32×32 images at 300 epochs and patch size 4. All experiments are performed on a single GPU.

| Model | Backbone | Parameters (M) | Flops (G) | Top-1 (%) | Top-5 (%) |
|-------|----------|----------------|-----------|-----------|-----------|
| GFN | ViT-XS | 6 | 0.38 | 70.91 | 88.57 |
| SA | ViT-XS | 9 | 0.58 | 62.00 | 83.71 |
| AFNO | ViT-XS | 7 | 0.46 | 70.60 | 90.10 |
| MWA | ViT-XS | 7 | 0.48 | **71.60** | 89.74 |
| GFN | ViT-S | 15 | 0.90 | 71.20 | 88.03 |
| SA | ViT-S | 21 | 1.40 | 61.90 | 82.83 |
| AFNO | ViT-S | 16 | 1.05 | 71.90 | 89.08 |
| MWA | ViT-S | 16 | 1.09 | **73.20** | 88.45 |
| GFN | ViT-B | 58 | 3.63 | 71.50 | 87.36 |
| SA | ViT-B | 85 | 5.52 | 62.20 | 89.32 |
| AFNO | ViT-B | 66 | 4.39 | 72.81 | 88.17 |
| MWA | ViT-B | 66 | 4.37 | **75.30** | 89.32 |

layers on the CIFAR10-100 dataset and a hidden size of 768. Also, all configurations use a token size of $4 \times 4$ to model sequence size settings. We use global average pooling at the last layer to produce output softmax probabilities for classification. We trained all models for 300 epochs with Adam optimizer and cross-entropy loss using a learning rate of $5 \times 10^{-4}$. We also use five epochs of linear learning-rate warm-up. We use a cosine decay schedule with a minimum value of $10^{-5}$, along with a smooth gradient cut-off to stabilize the training that does not exceed a value of 1, and the weight-decay regularization is set to 0.05. In particular, we use different transformer layers and adjust the hyperparameters of interest in AFNO and MWA to achieve a close and comparable number of parameters. More details about each model are provided below.

- SA uses 8 attention heads and a hidden size of 384 in the ViT-XS/4 and ViT-X/4 configuration and a hidden size of 768 in the ViT-B/4 configuration Dosovitskiy et al. (2020).

- GFN uses a hidden size of 384 in the ViT-XS/4 and ViT-X/4 configurations and a hidden size of 768 in the ViT-B/4 configuration Rao et al. (2021).

- The adaptive neural Fourier operator (AFNO) uses a hidden size of 384 in the ViT-XS/4 and ViT-S/4 configurations and a hidden size of 768 in the ViT-B/4 configuration. It also uses a scatter threshold of 0.1 in the ViT-B/4 and ViT-S/4 configuration and a scatter threshold of 0.01 in the ViT-XS/4 configuration (with a block count of 4-3 to reach the desired number of parameters) Guibas et al. (2021).

- MWA of 384 hidden size in ViT-XS/4 and ViT-S/4 configuration and 768 hidden size in ViT-B/4 configuration and 2D and 3D group convolution with kernel size 3 and 1 as possible weights. learning uses (along with the number of groups 6-8 to reach the desired number of parameters).

**Remark**: The ViT backbone used in our experiments is slightly different from the original ViT architecture. We use a token size of 4 compared to the token size of 16 used in the original ViT architecture. Also, MWA, AFNO, and GFN have geometric inductive biases which does not need a lot of data for learning. But SA has essentially no inductive biases and it is supposed to learn it from data. As a result, self-attention performs poorly on small datasets such as CIFAR without pretraining on larger datasets. Hence, we observe that self-attention performs poorly compared to the Fourier-based methods as well as MWA.

## 7.2 CIFAR CLASSIFICATION

We perform image classification experiments with the MWA mixer module and using the backbone ViT-XS/4, ViT-S/4 and ViT-B/4 on the CIFAR-10 and CIFAR-100 dataset containing 10,000 test sets with 10 and 100 classes, respectively, with resolution 32 × 32. We measure performance using top-1 and top-5 accuracy along with flops for different model parameters.

**CIFAR classification:** Classification results for different mixers are shown Table 2 and Table 3. It can be seen that the proposed MWA using DWT, can learn multiscale as well as non-periodic patterns

Table 3: Comparisons of different transformer-style architectures for image classification on CIFAR-10. All our models are trained on 32×32 images at 300 epochs and patch size 4. All experiments are performed on a single GPU.

| Model | Backbone | Parameters (M) | Flops(G) | Top-1(%) | Top-5(%) |
|-------|----------|----------------|----------|----------|----------|
| GFN | ViT-XS | 6 | 0.38 | 93.40 | 99.72 |
| SA | ViT-XS | 9 | 0.58 | 89.00 | 99.36 |
| AFNO | ViT-XS | 7 | 0.46 | 92.00 | 99.64 |
| MWA | ViT-XS | 7 | 0.48 | **94.30** | 99.75 |
| GFN | ViT-S | 15 | 0.90 | 94.40 | 99.69 |
| SA | ViT-S | 21 | 1.40 | 88.00 | 99.30 |
| AFNO | ViT-S | 16 | 1.05 | 93.70 | 99.67 |
| MWA | ViT-S | 16 | 1.095 | **95.30** | 99.70 |
| GFN | ViT-B | 58 | 3.63 | 95.30 | 99.58 |
| SA | ViT-B | 85 | 5.52 | 83.90 | 99.19 |
| AFNO | ViT-B | 66 | 4.39 | 95.20 | 99.59 |
| MWA | ViT-B | 66 | 4.37 | **96.10** | 99.60 |

Table 4: Comparisons of different transformer-style architectures for image classification on Tiny-ImageNet. All our models are trained on 64×64 images at 300 epochs and patch size 4. All experiments are performed on a single GPU.

| Model | Backbone | Parameters (M) | Flops (G) | Top-1 (%) | Top-5 (%) |
|-------|----------|----------------|-----------|-----------|-----------|
| GFN | ViT-XS | 7 | 1.52 | 60.87 | 82.03 |
| SA | ViT-XS | 9 | 2.53 | 46.50 | 70.41 |
| AFNO | ViT-XS | 7 | 1.80 | 59.30 | 81.55 |
| MWA | ViT-XS | 7 | 1.92 | **61.40** | 81.32 |
| SA | ViT-S | 21 | 6.07 | 46.70 | 69.08 |
| GFN | ViT-S | 16 | 3.64 | 61.20 | 80.32 |
| AFNO | ViT-S | 17 | 4.32 | 59.60 | 79.98 |
| MWA | ViT-S | 17 | 4.54 | **61.92** | 81.40 |

in the images better than the Fourier transform, which leads to higher than 1 accuracy improvements over existing Fourier-based mixers such as AFNO and GFN.

### 7.3 TINY-IMAGENET CLASSIFICATION

We perform image classification experiments with the MWA mixer module and using the backbone ViT-XS/4 and ViT-S/4 on the Tiny-ImageNet dataset that contains 100,000 images of 200 classes downsized to 64×64 colored images. Each class has 500 training images, 50 validation images and 50 test images. We measure performance through max top-1 and max top-5 validation accuracy along with flop and model parameters.

**Tiny-ImageNet classification:** Classification results for different mixers are shown in Table 4. It is observed that our proposed MWA, thanks to multiscale wavelet features that exist in natural images, outperforms global Fourier based methods including AFNO and GFN by more than 1 in top-1 accuracy. It also significantly outperforms SA when the patch size is chosen to be 4.

## 8 CONCLUSIONS

We introduced Multiscale Wavelet Attention (MWA) for transformers to effectively learn small-to-large range dependencies among the image pixels for representation learning. MWA adapts wavelet neural operators from PDEs and fluid mechanics after making basic corrections to WNO for natural images. MWA incurs linear complexity in the sequence size and enjoys fast algorithms for wavelet transform. Our experiments for image classification on CIFAR and ImageNet data show the superior accuracy of our proposed MWA block compared with alternative Fourier based attentions. There are still important directions to pursue. One of those pertains to more extensive evaluations with larger datasets and complex images involving multiscale features. Also, studying the performance of MWA for larger networks and data is an important next step that demands sufficient computational resources.

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

## A  APPENDIX

This section includes more experiments and ablations to analyze the performance of our proposed MWA.

### A.1  MWA WITH DIFFERENT DECOMPOSITION LEVELS

We perform image classification experiments using ViT-XS/4 and ViT-S/4 backbones on CIFAR-100 dataset, as well as ViT-XS/4 backbones on CIFAR-10 and Tiny-ImageNet datasets. We report the performance of these experiments through Top-1 and Top-5 accuracy, as well as the number of parameters and training time for the proposed MWA using different decomposition levels.

The classification accuracy of the proposed MWA with different decomposition levels is listed in Table A.1, Table A.1 and Table A.1. It can be seen that the proposed MWA achieves good accuracy in Top-1 at the decomposition level of m=1 due to the use of scale 1 bandpass coefficients that provide the best resolution. In addition, the results show that its training time is faster than other different decomposition levels; Because as the level of analysis increases in wavelet analysis, the number of coefficients to be processed also increases. This leads to additional calculations that can slow down the overall processing speed. At decomposition levels higher than one, due to the use of the coarsest band coefficients, the higher the decomposition level, the more details are obtained from the images, so the accuracy of Top-1 increases, but the training time becomes longer.

Table 5: Comparisons of proposed MWA with different decomposition levels on CIFAR10. All our models were trained with $32 \times 32$ images in 300 epochs (all experiments were performed on a single GPU).

| Model | Backbone | m | Top-1 (%) | Top-5 (%) | Train-Time |
|-------|----------|---|-----------|-----------|------------|
| MWA | ViT-XS | 1 | **94.30** | 99.75 | 4:25:00 |
| MWA | ViT-XS | 2 | 93.40 | 99.60 | 5:45:59 |
| MWA | ViT-XS | 3 | 93.52 | 99.63 | 6:16:47 |
| MWA | ViT-XS | 4 | 93.73 | 99.70 | 8:26:40 |

Table 6: Comparisons of proposed MWA with different decomposition levels on CIFAR100. All our models were trained with $32 \times 32$ images in 300 epochs (all experiments were performed on a single GPU).

| Model | Backbone | m | Top-1 (%) | Top-5 (%) | Train-Time |
|-------|----------|---|-----------|-----------|------------|
| MWA | ViT-XS | 1 | 71.60 | 89.74 | 4:36:12 |
| MWA | ViT-XS | 2 | 71.40 | 89.62 | 5:49:59 |
| MWA | ViT-XS | 3 | 72.30 | 89.93 | 6:26:07 |
| MWA | ViT-XS | 4 | 73.40 | 90.40 | 8:56:16 |
| MWA | ViT-S | 1 | 73.20 | 88.45 | 10:26:59 |
| MWA | ViT-S | 3 | **74.30** | 90.04 | 13:46:47 |

## A.2 MWA WITH DIFFERENT WAVELET TYPES

We perform image classification experiments using the ViT-XS/4 backbone on the CIFAR-100 dataset using the ViT-XS/4 backbone. We report the performance through Top-1, Top-5 accuracy and number of parameters along with training time for the proposed MWA with different wavelets.

The classification accuracy of the proposed MWA with different wavelets is listed in Table A.2. It can be seen that the training time using the Haar wavelet is faster than Daubechies, db4 and db6 wavelets because the Haar wavelet transform has a simpler structure and requires less calculations. The Haar wavelet uses only two coefficients for decomposition and reconstruction and It is good at detecting sudden changes in an image or signal, making it ideal for edge detection and image compression. The wavelets Daubechies use more coefficients for decomposition Daubechies (1992). This means that the computations required for the Haar wavelet transform are much simpler and faster compared to the computations required for the Daubechies transform. Also, the use of Daubechies wavelets is more suitable for data with complex and heterogeneous components.

## A.3 MWA WITH DIFFERENT NUMBER OF GROUPS

we conduct image classification experiments on CIFAR-100 dataset, utilizing ViT-XS/4 backbone. The proposed MWA approach with different number of groups was employ, and we reported the performance through Top-1 and Top-5 accuracy, number of parameters, and training time.

The classification accuracy of the proposed MWA with different number of groups is listed in Table A.3. When we increase the number of groups in convolution, we actually divide the input channel into separate groups and apply convolution to each group separately. This increases the number of independent computations required, which can slow down the overall process; Therefore, as the number of groups in the proposed architecture increases, the training time becomes longer, and as the number of groups decreases, the number of parameters increases and the training time decreases, but the computational complexity increases.

## A.4 MWA WITH DIFFERENT WAVELET TRANSFORMS

We perform image classification experiments using the ViT-XS/4 backbone on the CIFAR-100 dataset using the ViT-XS/4 backbone. We report the performance through Top-1, Top-5 accuracy and number of parameters along with training time for the proposed MWA with different wavelets.

The classification accuracy of the proposed MWA along with complex wavelet transform (DTCWT) and DWT wavelet transform are listed in Table A.4. Complex wavelet transform (DTCWT) is a type of wavelet transform commonly used in image processing applications Selesnick Ivan W. & Kingsbury (2005). The developed discrete wavelet transform is DWT. It decomposes the input into different wavelet coefficients at different resolutions. It uses two separate sets of filters to decom-

Table 7: Comparisons of proposed MWA with different decomposition levels on Tiny-ImageNet. All our models were trained with $64 \times 64$ images in 300 epochs (all experiments were performed on a single GPU).

| Model | Backbone | m | Top-1 (%) | Top-5 (%) | Train-Time |
|-------|----------|---|-----------|-----------|------------|
| MWA | ViT-XS | 1 | 61.25 | 81.32 | 22:32:45 |
| MWA | ViT-XS | 2 | 59.70 | 81.18 | 23:26:05 |
| MWA | ViT-XS | 3 | 60.10 | 81.12 | 24:18:10 |
| MWA | ViT-XS | 4 | 61.30 | 82.14 | 25:42:34 |
| MWA | ViT-XS | 5 | **61.80** | 82.04 | 26:36:40 |

Table 8: Comparisons of the proposed MWA with different wavelets on CIFAR100. All our models were trained with $32 \times 32$ images in 300 epochs (all experiments were performed on a single GPU).

| model | Backbone | Wave | Top-1(%) | Top-5(%) | Inference-Time | Train-Time |
|-------|----------|------|----------|----------|----------------|------------|
| MWA | ViT-XS | haar | 71.60 | 89.74 | 3.53 | 4:36:12 |
| MWA | ViT-XS | db6 | 71.60 | 89.06 | 6.85 | 8:36:31 |
| MWA | ViT-XS | db4 | 71.30 | 88.86 | 6.66 | 6:14:59 |

pose the input, one for the real part and one for the imaginary part, and instead of producing three output subbands like DWT, it produces 6 subbands. In addition, 6 subbands have real and imaginary outputs; Therefore, it obtains more coefficients than the discrete wavelet transform. Therefore, it increases the accuracy of Top-1 compared to DWT, but increases the training time because it involves additional calculations to improve the statistical properties of the wavelet coefficients.

A.5 MWA WITH DIFFERENT KERNEL SIZES

We perform image classification experiments using the ViT-XS/4 backbone on CIFAR-100 data. We report the performance through Top-1, Top-5 accuracy and number of parameters along with training time for the proposed MWA with different kernel size.

The classification accuracy of the proposed MWA with different kernel size is listed in Table A.5. It can be seen that as the size of the kernel increases, the training time becomes longer because when performing convolution, the size of the kernel determines the amount of computation required to process each input element. The larger the kernel size, the more elements of the input must be multiplied and summed by the kernel weight. Therefore, the number of required calculations increases. This can slow down the overall speed and lengthen the training time. Therefore, choosing the appropriate kernel size plays a crucial role in the performance of the proposed model.

A.6 COMPARISONS OF DIFFERENT TRANSFORMER STYLE ARCHITECTURES ALONG WITH TRAINING TIME

We perform image classification experiments using the ViT-XS/4 backbone on CIFAR-10 data. We report the performance through Top-1, Top-5 accuracy and number of parameters along with training time for different transformer style.

The Classification results for different mixers along with training time are shown table A.6. It can be seen that the GFN transformer is faster than AFNO and MWA transformers with group convolution due to its ability to effectively capture long-range dependencies in input sequences using global filter blocks.

A.7 COMPARISON OF DIFFERENT MODELS

We perform image classification experiments on CIFAR10 datasets. We report the performance through Top-1, Top-5 accuracy and number of parameters for different transformer style.

The Classification results for different mixers are shown in the Table A.7. It can be seen that the proposed MWA using DWT can analyze images at multiple scales, which makes it ideal for detecting features and patterns that occur at different sizes in the same image. resulting in improved accuracy top-1 Compared to ResNet and ResMLP.

Table 9: Comparison of proposed MWA with different number of groups on CIFAR100. All our models were trained with $32 \times 32$ images in 300 epochs (all experiments were performed on a single GPU).

| Model | Backbone | Params(M) | Groups | Top-1 (%) | Top-5 (%) | Train-Time |
|---|---|---|---|---|---|---|
| MWA | ViT-XS | 7 | 6 | 71.60 | 89.74 | 4:36:12 |
| MWA | ViT-XS | 13 | 1 | **73.03** | 89.98 | 4:09:25 |

Table 10: Comparisons of the proposed MWA with different wavelet transforms on CIFAR100. All our models were trained with $32 \times 32$ images in 300 epochs (all experiments were performed on a single GPU).

| Model | Backbone | Params(M) | Wavelet Transform | Top-1 (%) | Top-5 (%) | Train-Time |
|---|---|---|---|---|---|---|
| MWA | ViT-XS | 7 | DWT | 71.60 | 89.74 | 4:36:12 |
| MWA | ViT-XS | 7 | DTCWT | **72.20** | 89.30 | 9:45:54 |

Table 11: Comparisons of proposed MWA with different kernel sizes on CIFAR100. All our models were trained with $32 \times 32$ images for 300 epochs (all experiments were performed on a single GPU).

| Model | Backbone | Params(M) | kernal-size | Top-1 (%) | Top-5 (%) | Train-Time |
|---|---|---|---|---|---|---|
| MWA | ViT-XS | 7 | (1*1) | 61.81 | 85.52 | 4:01:39 |
| MWA | ViT-XS | 7 | (3*3) | 70.70 | 88.34 | 4:14:03 |
| MWA | ViT-XS | 7 | (3*3) (1*1) | **71.60** | 89.74 | 4:36:12 |
| MWA | ViT-XS | 7 | (5*5) (1*1) | 70.70 | 88.42 | 5:20:21 |
| MWA | ViT-XS | 7 | (5*5) (3*3) | 69.70 | 87.69 | 5:26:34 |

Table 12: Comparisons of different transformer-style architectures for image classification on CIFAR-10. All our models are trained on $32 \times 32$ images at 300 epochs and patch size 4. All experiments are performed on a single GPU.

| Model | Backbone | Top-1(%) | Top-5(%) | Inference-Time | Train-Time |
|---|---|---|---|---|---|
| MWA | ViT-XS | 94.30 | 99.75 | 3.78 | 4:25:00 |
| GFN | ViT-XS | 93.40 | 99.72 | 3.75 | 3:30:14 |
| AFNO | ViT-XS | 92.00 | 99.64 | 3.22 | 3:55:09 |

Table 13: Comparison of different models for image classification on CIFAR-10. All our models are trained on $32 \times 32$ images for 300 epochs. All experiments are performed on a single GPU.

| Model | FLop | Params(M) | Top-1(%) | Top-5(%) |
|---|---|---|---|---|
| MWA | 1.09 | 16 | **95.30** | **99.70** |
| ResMLP | 1.00 | 14 | 88.50 | 99.42 |
| ResNet-18 | 0.70 | 11 | 94.50 | 99.20 |

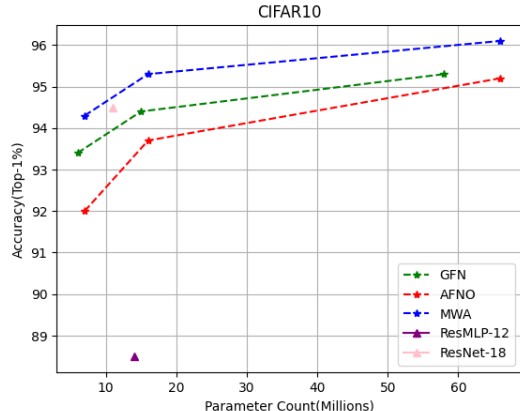

Figure 2: Diagram of Top-1 accuracy by number of parameters for classification on CIFAR10 dataset. As shown in the figure, the proposed MWA model performs better than other models.

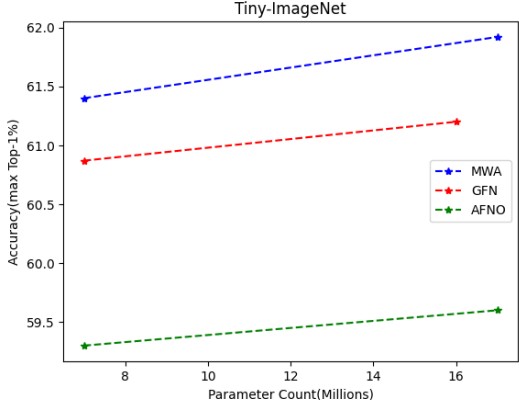

Figure 3: diagram of Top-1 accuracy by number of parameters for classification on Tiny-ImageNet dataset. As shown in the figure, the proposed MWA model performs better than other models based on Fourier transform.

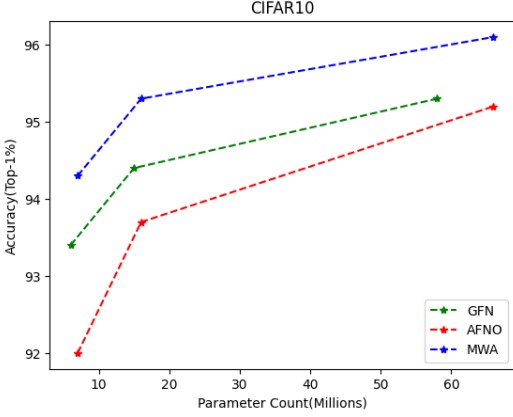

Figure 4: diagram of Top-1 accuracy by number of parameters for classification on CIFAR10 dataset. As shown in the figure, the proposed MWA model performs better than other models based on Fourier transform.

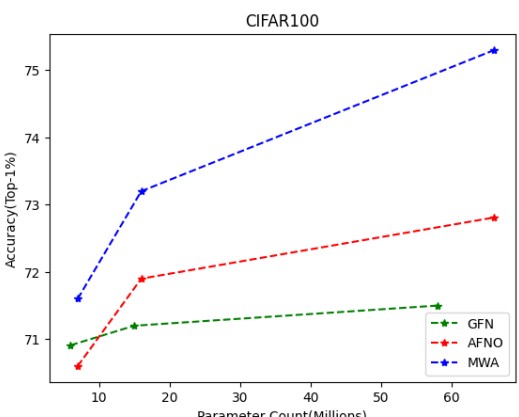

Figure 5: diagram of Top-1 accuracy by number of parameters for classification on CIFAR100 dataset. As shown in the figure, the proposed MWA model performs better than other models based on Fourier transform.

