# OpenReview forum: "MULTISCALE ATTENTION VIA WAVELET NEURAL OPERATORS FOR VISION TRANSFORMER"
_ICLR.cc/2024/Conference — Submitted to ICLR 2024_

### Official Review · Reviewer_UUhz · 2023-10-26

**Soundness:** 2 fair
**Presentation:** 2 fair
**Contribution:** 2 fair
**Rating:** 3
**Confidence:** 4

**Summary:**

The complexity of Transformer's algorithm grows quadratically with the length of the sequence, and an adaptive Fourier neural operator is introduced to construct a fully convolutional attention mechanism for high-resolution feature extraction. The author addresses the problem of neglecting fine and medium-scale structures in natural images in the common adaptive Fourier neural operator by employing a designed multi-scale wavelet attention to learn multi-scale features.They evaluated their method on two different publicly available datasets, whereas they mention that their method showed superior performance in comparison with existing methods of the literature.

**Strengths:**

Motivation and Meaning: motivation is feasible
In order to overcome the issue of adaptive Fourier neural operators ignoring common fine and medium-scale structures in natural images, they proposed a new wavelet neural operator.

Ablation experiment: adequate experimental content

Results：significant performance improvement
In comparison with existing methods, the proposed method has achieved superior performance under the same computational resources.

**Weaknesses:**

Innovation: the contribution not enough for a ICLR paper
Adaptive Fourier neural operators in classical work can mix the input tokens into continuous global convolution without any dependence on the input resolution, which is a huge innovative step. However, the multiscale wavelet attention presented in this paper fuses different scale features through a parallel structure of discrete wavelet transform and convolution. This one structural innovation alone is not enough to support the whole paper.

Method Reproducibility: no open source
The methodology section lacks specificity. It would be better to provide specific implementation details of the code.

Comparative experiment: inadequate comparison of methods
The comparative experiments are not comprehensive enough. It is suggested to provide additional comparative experiments on tasks such as segmentation or detection.


Written expression: average expression in writing
Poor graphic layout, logical approach to the article, inadequate experimental analysis

**Questions:**

1.Why MWA's Top-5% accuracy in Table 2 is not optimal？

2.Fusion strategy for summing parallel features in MWA, why is it not concate？

3.Is the improvement in the performance of the proposed method due to this parallel structure of fusing features at different scales, or is it the discrete wavelet transform?

---

> ### Author Response · Authors · 2023-11-20
> **Response to Reviewer UUhz**
>
> Thank you for your constructive comments and suggestions, and they are exceedingly helpful for us to improve our paper. We have carefully incorporated them in the revised paper. In the following, your comments are first stated and then followed by our point-by-point responses.
>
> Q1: Why MWA's Top-5% accuracy in Table 2 is not optimal？
> top-5% accuracy refers to the accuracy achieved by the model on the most difficult or challenging samples in the dataset.While the top-5% accuracy is indeed important and can provide insights into the model's performance on difficult cases, it may not reflect the overall performance of the model on the entire dataset. It is possible that the model performs better on the majority of the samples, but encounters some particularly challenging cases where its accuracy drops.Optimizing for top-5% accuracy alone not be the primary goal. Top 1 is more important.
>
> Q2: Is the improvement in the performance of the proposed method due to this parallel structure of fusing features at different scales, or is it the discrete wavelet transform?
> both

---

### Official Review · Reviewer_5HJK · 2023-10-30

**Soundness:** 2 fair
**Presentation:** 2 fair
**Contribution:** 2 fair
**Rating:** 3
**Confidence:** 4

**Summary:**

This paper proposes a new efficient self-attention for vision transformer to reduce the quadratic cost of standard self-attention. By exploiting the multi scale structures in natural images, the authors introduce a multi-scale wavelet attention, which perform of a convolution with nonlinearity on the wavelet coefficients of input feature maps and followed by an inverse wavelet transform. Additionally, two convolutional skip branches are added to help learning high frequency details. The experiments show that the proposed method performs better compared to other baselines.

**Strengths:**

1. The proposed method introduces a multi-resolution inductive bias, which is useful for natural images.

2. The authors performed extensive ablation study on different configurations of the proposed method to understand the behavior of the proposed idea.

**Weaknesses:**

1. Section 4 seems unnecessary for introducing the proposed idea. It might be better to reduce the text length and move to the related work.

2. Multi-resolution based attentions have been proposed in multiple papers, such as H-Transformer (https://arxiv.org/abs/2107.11906) and MRA Attention (https://arxiv.org/abs/2207.10284). Since this paper also proposes multi-resolution attention, it should have a discussion on these works.

3. It would be better to have an ablation study to evaluate the performance of three cases: 1. Three branches in Figure 1 are all enabled. 2. Only multi-scale branch is enabled. 3. Only two convolutional branches are enabled. This ablation study allows isolating the effects of two convolutional branches from the effects of multi-scale branch to better evaluate the multi-scale attention.

4. Experiments are only performed on small datasets (CIFAR and Tiny-ImageNet). It might be difficult to generalize the conclusion to larger scale datasets (such as ImageNet).

5. The motivation of this paper is to solve the quadratic complexity of self-attention for high resolution vision, but all experiments are performed on short sequences: 64 tokens (32 x 32 with patch size 4) or 256 tokens (64 x 64 with patch size 4).

6. The multi scale attention is operating on the (8 x 8 or 16 x 16) feature maps of the image. It is not clear whether the model really needs the multi-scale inductive bias at this scale.

**Questions:**

1. The reference links for tables in Appendix seems to be broken.

---

> ### Author Response · Authors · 2023-11-20
> **Response to Reviewer 5HJK**
>
> Thank you for your constructive comments and suggestions, and they are exceedingly helpful for us to improve our paper. We have carefully incorporated them in the revised paper. In the following, your comments are first stated and then followed by our point-by-point responses.
>
>  Due to the GPU limitation, we could not perform our experiments on large data  and large pach size  (all experiments were performed on 1 GPU), but we believe that our model performs better than other models on large data.

---

### Official Review · Reviewer_EE4x · 2023-10-31

**Soundness:** 3 good
**Presentation:** 2 fair
**Contribution:** 3 good
**Rating:** 6
**Confidence:** 3

**Summary:**

This study tries to address the limitations of the Adaptive Feedforward Neural Oscillator (AFNO). The authors suggest the utilization of the Haar wavelet transform, a well-established technique in image processing renowned for its efficacy in providing a multiscale representation of natural images. Researchers use a changed version of the wavelet neural operator to learn about a multiscale mixer. This operator has been studied a lot for how well it solves partial differential equations (PDEs) in the field of fluid mechanics. The design is adjusted to accommodate high-resolution natural images that have discontinuities resulting from objects and edge features.

**Strengths:**

They use MWA in place of the attention in ViT.
Show a substantial improvement over other Fourier-based attentions like global filter network (GFN) and AFNO.

**Weaknesses:**

More datasets need to be evaluated

Large datasets like ImageNet on classification task

**Questions:**

Is there a reason why the method was not applied for other large datasets or other tasks?

---

> ### Author Response · Authors · 2023-11-20
> **Response to Reviewer EE4x**
>
> Thank you for your constructive comments and suggestions, and they are exceedingly helpful for us to improve our paper. We have carefully incorporated them in the revised paper. In the following, your comments are first stated and then followed by our point-by-point responses.
>
> Q1: s there a reason why the method was not applied for other large datasets or other tasks?
> Due to the GPU limitation, we could not perform our experiments on large data (all experiments were performed on 1 GPU), but we believe that our model performs better than other models on large data.

---

### Official Review · Reviewer_JC9v · 2023-11-03

**Soundness:** 4 excellent
**Presentation:** 1 poor
**Contribution:** 3 good
**Rating:** 3
**Confidence:** 4

**Summary:**

This paper proposes Multiscale Wavelet Attention (MWA) an alternative to self-attention that relies on the wavelet transform to leverage coarse-to-fine scale structures in their processing structure. The proposed MEA demonstrates improvements over Fourier-based attention methods such as AFNO, GFN and normal self-attention (SA) on CIFAR and Tiny ImageNet.

**Strengths:**

To the best of my knowledge, this paper introduces the interesting and novel idea of using the Wavelet Transform within a Transformer. The proposed method is backed up experimentally in small and medium-sized datasets.

**Weaknesses:**

The main weakness of this work is given by its presentation. While I believe the idea presented here to be promissing, I find the way in which it has been presented to be too convoluted, with motivations as well as many important details missing. In addition, the writing seems rushed, which is highlighted by replicated parts in the paper as well as some imprecisions in definitions and claims. Specifically:

**Preliminaries**

- “... self-attention mechanism, an inductive bias that associates …” -> Do you mean operation? To the best of my understanding, an inductive bias is something different.

- “.. is a global operator, which means it may overlook or miss fine and moderate scale structures commonly present in natural images.” -> I do not agree this is the case in general. To the best of my understanding, this is only the case if the global kernel used is intrinsically low-frequent. To give an example, several modern sequential models that rely on global convolutions –either parameterized as Implicit Neural Representations [1,2,3,4] or SSMs [5,6,7,8]– are global yet able to describe high-frequencies. This ability allows them to model long sequences such as Language [9] and DNA sequences [10] in a fine-grained manner. If your claim were true, this would not be possible –An analysis about this is given in [1].

- Furthermore, note that these parameterizations also allow for the constructions of CNNs that require minimal inductive biases for various designs, e.g., [11, 12] –see first paragraph.

- Related to the previous comments: Note that long convolutional models, e.g., [1 - 11] all rely on smart parameterizations of convolutional kernels. Hence, it is not true that all global convolutional models have a large number of parameters for high resolution inputs  –see Fourier based attention in the Related Work.

- The authors present the non-adaptability of methods like MLP mixers as a limitation. However, to the best of my understanding, MWA also is non-adaptable –see MLP based attention in the Related Work.

- “FFT can extract periodic patterns due to the use of sine and cosine functions to analyze images, but it is not suitable for studying the spatial behavior of images with non-periodic patterns.” -> I believe this is incorrect. Do note that there exist several theorems that state that the Fourier transform can be used to describe arbitrary functions on a finite domain. If this were a problem, compression methods such as JPEG would not be as powerful and successful as they are. There is something to be said about periodicity, but just as for the Wavelet Transform, this comes from the assumption that signals are periodic outside the domain considered. Note that, like the Fourier transform (and FFT), applying convolutions on the Wavelet domain, also has a form of “circular convolution”, which relies on the exact same assumption of periodicity.

- In Table 1, I am not sure I understand where the quadratic terms of MWA and AFNO come from. If this were the case, then FFT convolutions as well as “Wavelet convolutions” would also have quadratic complexity, right?

**Method.**

- “In this work, we use all the coefficients of the last decomposition level”. What do you mean here? Do you mean that you use all the coefficients up to that level? or only the ones of the last level of decompositions? Also, in practice only 1 level of decomposition is used. It is difficult to understand what this is referring to.

- Related to the previous comment: “In DWT, the wavelet coefficients in the highest scale include the important features of the input…” I am not sure I understand what is meant here.

- The author defines the Wavelet Neural Operator as a convolution on the Wavelet domain by means of multiplication. However, in Eq. 10, the authors suddenly change the operation to convolution on this space. Aside from not being motivated anywhere, this makes the method unnecessarily difficult to understand. Given that a convolution in Wavelet space is equivalent to multiplication in the spatial domain, I do not understand why this is used here. What is the motivation of using convolution on that space instead of multiplication?

- In the 4th step, the authors introduce residual connections which are actually not residual. These are simply other branches composed of convs and non-linearities.

**Empirical Setup.**

- It is not clear from the paper what kind of “Conv” is used in the Wavelet space –I also went through the appendix and I was unable to find this.

- Also, is the Wavelet decomposition used as an spatial input, or is each decomposition considered independently from one another? How are frequency components “mixed” across levels of decomposition?

**Evaluation.**

- Most of the papers’ motivation swirls around efficiency. However, the authors do not evaluate the approach on large scale datasets such as ImageNet. Is there a reason for this? In my opinion it is difficult to evaluate the effect of using multi-scale approaches on low-resolution data. Is this the reason only a single level of decomposition is used?  Empirical evaluations on higher resolution datasets would be more convincing given this fact.

- “... It also significantly outperforms SA when the patch size is chosen to be 4.” -> Which other patches were analyzed? I could not find any details about this.

- It would be good to have a comparison in terms of time with other methods. Specifically, if I understood the paper well, I would expect the running time to be slower than methods based on global convolutions.

- In Experiments, the authors refer to the patch size as “token size”. I would strongly encourage the authors to use consistent nomenclature throughout the paper.

**Repetitions.**

One of the items at the end of Sec 5 is repeated. The definition of the datasets used in the experiments section is also repeated.

**Appendix.**

- The labeling of the tables and images here is incorrect.

**Minor**

- Please make correct use of \pcite and \tcite

- Tabel is a consistent typo throughout the paper. The right word is Table.

**Questions:**

[see Weaknesses section].

### Conclusion

Despite the fact that I believe the idea to be interesting, given the previous observations, I do not believe the paper is ready for publication. I would encourage the authors to consider rewriting the paper and motivating the method well. Furthermore, I would encourage the authors to carry out experiments in higher frequency images, as this is the setting on which MWA has the potential to really shine.

---

> ### Author Response · Authors · 2023-11-20
> **Response to Reviewer JC9v**
>
> Thank you for your constructive comments and suggestions, and they are exceedingly helpful for us to improve our paper.
>
> Q1: ... self-attention mechanism, …” .
> In the context of self-attention mechanisms, "bias" is actually the correct term, not "operation". The phrase should be "an inductive bias that associates ...". The attention mechanism introduces a bias towards certain interactions or associations between elements.
>
> Q2 :  “.. is a global operator, ....
> AFNO, as a global operator, may sometimes overlook or miss fine and moderate scale structures commonly found in natural images. It does not make a general claim about all global operators. The effectiveness of global operators in capturing fine and moderate scale structures depends on the characteristics and design of the specific operator or model used. Modern sequential models, like the Implicit Neural Representations (INRs) and Structured-Sparse Models (SSMs) you mentioned, have demonstrated the ability to model high frequencies and capture fine-grained details even with global convolutions.These models utilize parameterizations and structures that enable them to capture and represent both low and high-frequency components adequately. This allows them to describe long sequences, such as language and DNA sequences, in a fine-grained manner.Therefore, it is essential to consider the specific characteristics and design choices of the global operator or model being used before making broad statements about their ability to capture fine and moderate scale structures.
>
> Q3 : “FFT can extract periodic patterns.....
>
> The Fourier Transform, including its discrete version (FFT), is indeed a powerful tool for analyzing signals and images, not limited to periodic patterns. It can represent arbitrary functions in terms of sinusoidal components, and its application extends to a wide range of signal processing tasks, including image analysis.While it's true that Fourier analysis is particularly well-suited for periodic signals due to the use of sine and cosine functions, it doesn't mean it's limited to periodic patterns. In fact, as you pointed out, the Fourier Transform can be applied to describe arbitrary functions on a finite domain. But Wavelet transform works better than Fourier transform for the following reasons
> 1. Time-Frequency Localization: Wavelets provide excellent time-frequency localization, which means they can capture both high-frequency and low-frequency details of a signal at different time points. This is particularly useful when analyzing signals with non-stationary characteristics, where frequency content changes over time.
> 2. Multiresolution Analysis: Wavelets offer a multiresolution analysis, meaning that they can capture signal details at different scales. This allows for the identification of localized features within a signal. Fourier transforms, on the other hand, provide a global frequency representation, making it difficult to analyze localized features.
> 3. Edge Detection: Wavelets are effective in detecting edges and discontinuities in signals due to their ability to capture
>
> Q4: “In this work, we use all the coefficients .....
> All the coefficients of the last level of analysis. We also did it with higher levels Decomposition.
>
> Q5:  “In DWT, the wavelet coefficients.....
> In DWT, the wavelet coefficients in the highest scale refer to the smallest details or features in the input. These details are considered important as they contribute to the overall understanding of the input.
>
> Q6 : The author defines the Wavelet Neural Operator as a convolution ....
>
> Convolution operates by sliding a small kernel over the input space, multiplying the kernel's values with the corresponding input values at each position, and summing them up. This process takes into account the neighboring values, allowing for the detection of patterns, edges, and other spatially dependent features.Multiplication, on the other hand, is a point-wise operation that does not consider the surrounding values. It is mainly used for scalar multiplication or element-wise operations, which may not capture the spatial relationships present in the data.By using convolution, we can effectively capture spatial information and exploit the locality of features in the input space, making it especially useful in tasks like image processing, where recognizing patterns and extracting relevant features require considering the context and relationships between neighboring pixels.
>
> Q8 :  is the Wavelet decomposition used as an spatial input, or is each decomposition considered independently from one another? How are frequency components “mixed” across levels of decomposition?
>
> In wavelet decomposition, the input signal is divided into different frequency bands or levels. Decomposition is done recursively, different frequency components are not mixed in each level. Instead, they are analyzed independently.

---

### Official Review · Reviewer_Thgr · 2023-11-04

**Soundness:** 3 good
**Presentation:** 2 fair
**Contribution:** 1 poor
**Rating:** 3
**Confidence:** 4

**Summary:**

The paper introduces Multiscale Wavelet Attention (MWA) to make token mixing in image transformers more efficient and better capture correlations at different scales. The research focuses on the ViT architecture and modifies the self-attention block. The authors use a wavelet neural operator to perform token mixing of different sizes.

The experiments demonstrate that the proposed architecture outperforms the Adaptive Fourier Neural Operator (AFNO) and several other transformer-based models on CIFAR-10/100 and TinyImageNet.


### Final Decision
After reading the other reviews and the answers of the authors, I have to lower my rating. Thanks to the other reviews, I better understood the limitations of the presented approach.

**Strengths:**

- The paper is well-written and easy to follow. The main contributions are clearly stated.
- The motivation for using a wavelet-based modification is clear, and the cited connections to previous research in this field make it easier for a reader to understand the authors' ideas.
- The theory is well explained and appears to be a straightforward extension of the AFNO idea to another operator.
- The experiments demonstrate that the improvement over AFNO is not marginal.

**Weaknesses:**

The main weakness of the paper is the fact that there is no experimental justification, that the improvement is caused by a better analysis of multiscale features. It is hard to tell from the provided experimental results, that the improived performance is a result of better mixing of tokens of different scales. One may assume that the improvement is a result of a better computational graph which, due to the stochastich nature of optimization, by accident leads to a better performance.

Minor weakness of the paper is the formatting, which requires several improvements:

1. Tables: Ensure that all numbers use the same precision, i.e., the same number of decimal places. For instance, in Table 2, Row Top1%, there should not be 62 and 72.81; it should be 62.0 and 72.8, or if you wish, 62.00 and 72.81. In that case, the rest of the numbers should also have two decimal places.
2. Figure 1: The diagram at the bottom appears to be of rather low quality. It should be drawn in a more consistent way with the rest of the image. The same applies to the right diagram.

**Questions:**

- Can you design an experiment which will demonstrate that the improved results are indeed caused by a better analysis of tokens of different scales?
- What are the hyperparameters of the method? Can you perform ablation studies on them?

---

> ### Author Response · Authors · 2023-11-21
> **Response to Reviewer Thgr**
>
> Thank you for your constructive comments and suggestions, and they are exceedingly helpful for us to improve our paper. We have carefully incorporated them in the revised paper. In the following, your comments are first stated and then followed by our point-by-point responses.
>
> Q1: Can you design an experiment which will demonstrate that the improved results are indeed caused by a better analysis of tokens of different scales?
> Yes, the expriments can be shown on big data, but it requires gpu. We did all our expriments with 1 gpu.
>
> Q2: What are the hyperparameters of the method? Can you perform ablation studies on them
>
> Some common hyperparameters could include the number of decomposition levels, wavelet function,  kernel size...
> Yes, to perform ablation studies on these hyperparameters, one can systematically change one hyperparameter at a time while keeping the others constant.

---

### Meta-Review · Area_Chair_3o9k · 2023-12-05

**Metareview:**

The paper introduces Multiscale Wavelet Attention (MWA)  for vision transformer to reduce the quadratic cost of self-attention and also better capture features at different scales. It performs of a convolution with nonlinearity on the wavelet coefficients of input feature maps and followed by an inverse wavelet transform. Experimental results show that MWA outperforms baselines.

The main strength of this work is that its proposed Multiscale Wavelet Attention can reduce the cost and also efficiently learn multiscale features. However, three reviewer emphasizes the insufficient experiments (e.g., no experiments on large-scale datasets), the poor writing, and possible reproduction  in this work. I partly agree with the reviewer's comments, especially for the experiments. So to improve this work, the authors can revise it to well introduce the motivation, methods and also important details, and also evaluate the proposed method on large-scale dataset, e.g., the widely used imagenet dataset.

Considering most reviewers have low intentions to accept this work, we cannot accept it. Note that one reviewer who gave a score of 6 did not provide any detailed comments, his score has a much lower weight. Accordingly, we will not accept this work.

**Justification For Why Not Higher Score:**

For five reviewers, three reviewers give the score of 3, and think that this work has several issues, including the insufficient experiments (e.g., no experiments on large-scale datasets), the poor writing, and possible reproduction in this work. I partly agree with the reviewer's comments, especially for the experiments.

Although the remaining two reviewers give score of both 6, one of them did not provide any detailed comments and his score has a much lower weight.

**Justification For Why Not Lower Score:**

N/A

---

### Decision · Program_Chairs · 2024-01-16

Reject